# Flavonoids in Skin Senescence Prevention and Treatment

**DOI:** 10.3390/ijms22136814

**Published:** 2021-06-25

**Authors:** Anna Domaszewska-Szostek, Monika Puzianowska-Kuźnicka, Alina Kuryłowicz

**Affiliations:** 1Department of Human Epigenetics, Mossakowski Medical Research Centre PAS, 02-106 Warsaw, Poland; mpuzianowska@imdik.pan.pl; 2Department of Geriatrics and Gerontology, Medical Centre of Postgraduate Education, 01-826 Warsaw, Poland

**Keywords:** senescent cells, senescence-associated secretory phenotype (SASP), flavonoids, senolytics, senostatics

## Abstract

Skin aging is associated with the accumulation of senescent cells and is related to many pathological changes, including decreased protection against pathogens, increased susceptibility to irritation, delayed wound healing, and increased cancer susceptibility. Senescent cells secrete a specific set of pro-inflammatory mediators, referred to as a senescence-associated secretory phenotype (SASP), which can cause profound changes in tissue structure and function. Thus, drugs that selectively eliminate senescent cells (senolytics) or neutralize SASP (senostatics) represent an attractive therapeutic strategy for age-associated skin deterioration. There is growing evidence that plant-derived compounds (flavonoids) can slow down or even prevent aging-associated deterioration of skin appearance and function by targeting cellular pathways crucial for regulating cellular senescence and SASP. This review summarizes the senostatic and senolytic potential of flavonoids in the context of preventing skin aging.

## 1. Introduction

Besides being an economic and social problem, aging is predominantly a medical issue. Thus, there is an increasing need to understand the mechanisms underlying this highly complex process [1], which inevitably leads to impaired body homeostasis and function, an increased risk of complex diseases, and, finally, death.

Cellular senescence contributes to age-related tissue and organ dysfunction and diseases through mechanisms that perturb stem cell niches, induce aberrant cell differentiation, disrupt the extracellular matrix, stimulate tissue inflammation, and induce senescence in neighboring cells [2,3,4]. Senescent cells secrete a specific set of pro-inflammatory cytokines, chemokines, growth factors, lipids, and proteases, a phenomenon called the senescence-associated secretory phenotype (SASP) [5]. It is believed that the accumulation of senescent cells in tissues contributes to the impairment of their homeostasis and increases the risk of many age-related diseases [6]. SASP, in turn, can lead to chronic inflammation (e.g., local or generalized) and changes in tissue structure and function [7]. Therefore, eliminating senescent cells or neutralizing SASP components may provide beneficial effects not only for the affected tissue but also the whole organism. Drugs that selectively eliminate senescent cells (senolytics) or neutralize SASP (senostatics) represent an attractive therapeutic strategy for delaying aging and age-related diseases [8].

Skin aging is associated with an increasing number of senescent cells and is related to many pathological changes, including decreased protection against pathogens, increased susceptibility to irritation, delayed wound healing, and increased cancer susceptibility [9]. Therefore, therapies that reduce senescent cell numbers or block SASP may be an effective treatment for aging-associated skin deterioration [10]. The senolytic and senostatic activities of several drugs (e.g., metformin and rapamycin) have already been demonstrated in preliminary clinical trials [11,12]. However, in vitro and in vivo data show that different flavonoids have similar properties; therefore, they can be considered a therapeutic option for skin aging prevention and treatment.

## 2. Skin Aging and Senescence

The skin consists of an outer epidermal layer (epidermis), which constitutes a barrier to the environment, and an inner dermal layer (dermis) connected by the basement membrane. The epidermis consists of a multi-layered epithelium containing mainly keratinocytes that proliferate from stem cells in the basal layer attached to the basement membrane. Subsequently, they detach, stop proliferating, and undergo a terminal differentiation program that ends in a specialized form of programmed cell death, known as cornification. The epidermis also contains melanocytes that protect against ultraviolet (UV) radiation due to their pigment content. Langerhans cells are a third cell type in the epidermis that belongs to the antigen-presenting dendritic cells. Epidermal homeostasis relies on the proper function and interactions of all these cellular components [13]. The dermis consists of the papillary layer just below the epidermal basement membrane and the lower reticular layer. The papillary layer contains fibroblasts, a small number of fat cells (adipocytes), blood vessels, and phagocytes, while the reticular layer contains fewer fibroblasts but thicker collagen fibers in the dermal matrix. The dermis is also comprised of nerve endings, vessels, pericytes, and cells of the immune system, including mast cells and macrophages [14].

Skin aging can be defined as intrinsic or extrinsic. Intrinsic skin aging is chronological and depends on endogenous factors, such as genetics and metabolic and hormonal status. Extrinsic skin aging is caused by environmental factors. Both intrinsic and extrinsic aging of the skin is caused by a disruption of gene expression, a decline in the recycling of defective mitochondria, and the accumulation of cellular by-products that lead to decreased cellular bioenergy [15,16].

During chronological aging, senescent cells accumulate in the dermis and epidermis. This accumulation can be induced and accelerated by various cellular perturbations, including DNA damage and mitochondrial dysfunction [17]. Several external factors, such as DNA damaging agents (e.g., X-rays, UV, and cigarette smoke), can induce senescence in the epidermis and dermis. UV radiation plays a central role in skin senescence and skin cancer development. UV radiation is composed of three main components based on photon wavelength: UVA having the longest wavelengths (315–400 nm), UVB being mid-range (290–320 nm), and UVC being the shortest wavelengths (100–280 nm). All UV types can act as environmental mutagens leading to direct and indirect (via increased production of oxidative free radicals) DNA damage, and each can result in mutagenesis in skin cells. UVA radiation is the most prevalent component of solar UV radiation. It penetrates deeper than UVB (that has a major action on the epidermis) into the skin and induces profound alterations of the dermal connective tissue [18,19]. In vitro studies also show that UVC has a deteriorating effect on genome stability, contributing to the aging of fibroblasts and keratinocytes [20,21]. However, considering that most of this radiation is absorbed by an ozone layer, its clinical relevance is less pronounced. To give the complete picture, it is also important to mention the effects of infrared radiation (IR) on skin aging. Recent studies indicate that IR and heat may induce premature skin aging by stimulation of matrix metalloproteinases (MMP) expression and modulation of elastin and fibrillin synthesis. Moreover, in human skin, heat stimulates the formation of new vessels, recruitment of inflammatory cells, and causes oxidative DNA damage [22].

The senescent cells in the skin can be identified by elevated expression of the cell-cycle inhibitors p21 and p16 and proteins involved in DNA repair, increased lysosomal enzyme β-galactosidase activity, loss of nuclear high mobility group box 1 (HMGB1), reduced lamin B1 expression, and chromatin remodeling [16,18].

Senescence is also manifested by a change in the cells secretory profile, such as increased secretion of interleukin (IL)-1α, IL-1β, IL-6, IL-8, MMP-1, and -3 that degrade the dermal matrix, and various growth and transcription factors [23]. Skin irradiation plays a central role in modulation of SASP, too. While most of UVC is blocked by the ozone layer, the UVA and UVB contribute to skin senescence and inflammation by activating SASP genes like IL-1, IL-6, and MMPs [24]. In turn, both UVA and UVB can downregulate tumour growth factor (TGF)-β, resulting in reduced collagen type I synthesis, leading to dermal thinning and wrinkle formation [25].

These hallmarks of senescence apply to multiple cell types in the skin; however, cells residing longer in the tissue are affected more severely by loss of cellular maintenance and repair mechanisms than those that are highly proliferative and replaced frequently [26]. The phenomenon of senescence affects all elements of the skin.

### 2.1. Keratinocytes

Once differentiated, keratinocytes leave the basal layer of the epidermis. At that point, they cannot proliferate and display some changes in cellular metabolism and chromatin rearrangements typical of senescent cells. However, the current consensus of the International Cell Senescence Association (ICSA) states that terminal differentiation of cells does not qualify them as senescent cells because the process of differentiation is not a result of stress or damage [27]. These cells lack some typical features of senescent cells, such as macromolecular damage, protein oxidation, telomere shortening, and SASP.

The process of keratinocyte senescence is complex and still under investigation. In vitro studies suggest that keratinocytes develop a senescent phenotype while lacking terminal differentiation markers [28]. Cellular availability of nicotinamide adenine dinucleotide (NAD) seems to be a critical factor in regulating this process. High levels of NAM (nicotinamide), the main precursor of NAD, inhibit differentiation of the upper epidermal layers and maintain proliferation in the basal layer. Preventing the conversion of NAM to NAD leads to premature differentiation of human primary keratinocytes and senescence [29].

Another feature of senescent keratinocytes is an accumulation of redox stress-induced single-strand DNA breaks that remain unrepaired due to a decrease in poly-ADP-ribosyltransferase (PARP1) activity and promote cell cycle arrest [30]. Senescent keratinocytes are also characterized by lower insulin growth factor receptor (IGF-1R) levels, resulting in impaired DNA damage responses [31]. Collagen 17A1 (Col17a1) appears to play an essential role in epidermal stem cell aging in vivo. Its depletion stimulates the terminal differentiation of aged keratinocytes, resulting in corneocyte formation [32]. Moreover, loss of Col17a1 in the epidermal basal keratinocytes disturbs the epidermal-dermal junction [29].

These keratinocyte changes can be accelerated by both UVA and UVB radiation; therefore, UV exposure seems to be the leading stimulus of keratinocyte senescence [33]. Because keratinocyte proliferation is the primary mechanism contributing to the renewal of the epidermis, the accumulation of nonproliferating senescent epidermal cells and prolonged exposure to senescent cell-related SASP cause disturbances in the regeneration of the epidermis of older individuals and contribute to the development of neoplasia and impaired wound healing [34].

### 2.2. Fibroblasts

Fibroblasts are the most abundant cells of the dermis, and their dysfunction significantly contributes to skin aging. The main features of fibroblast senescence include accumulation of double-strand DNA breaks, oxidative DNA damage, chromosomal and epigenetic aberrations, shortening or oxidation of telomeres, and the impairment of DNA repair mechanisms. Another feature of fibroblast senescence is the loss of cellular proteome homeostasis that manifests as aberrant synthesis; post-translational modifications; degradation of proteins; and changes in the synthesis and secretion of lipids, nucleic acids, and other metabolites. In human skin aging, senescent fibroblasts mainly accumulate in the dermis. Compared to non-senescent cells, senescent fibroblasts are characterized by a reduced extracellular matrix and increased MMP production. Interestingly, senescent skin fibroblasts can transfer extracellular vesicles (EV) containing bioactive microRNAs and SASP components to cells in spatial proximity (e.g., keratinocytes) to spread their senescent features [35]. In contrast to keratinocytes, UVA radiation due to its deeper penetration is the chief stimulus inducing fibroblast senescence in vivo [18,19], while all types of UV radiation and X-rays have been shown to stimulate fibroblasts senescence in vitro [36,37].

### 2.3. Melanocytes

Even though melanocytes constitute 5–10% of the cells in the basal layer of the epidermis, they significantly impact skin aging.

Melanocytes contain specialized lysosome-lineage organelles called melanosomes dedicated to synthesis and storage of melanin, a photoprotective pigment that protects skin from UVB, UVA, and visible blue light. Melanin-containing melanosomes can be transferred from melanocytes to the surrounding keratinocytes that together constitute a melano-epidermal unit. Melanin acts as a redox UV-absorbing agent and, in this way, directly prevents the DNA of epidermal cells from photodamage. However, melanin contributes to DNA protection also indirectly by scavenging reactive oxygen species (ROS) formed during the UV-inducing oxidative stress in the skin [38]. Aging is associated with several changes in the skin pigmentary system that can be accelerated by exposition to UV radiation, leading to structural changes in melanocytes and their hyperactivity. Ectopic up-regulation of melanocytes contributes to the formation of senile lentigines/lentigo and other age-related hyperpigmentation disorders and may result in the development of melanoma—the most lethal of all types of skin cancers—in which incidence grows with age [39].

Moreover, it was shown that the medium from senescent melanocytes caused a decrease in fibroblast proliferation when added to fibroblast cell culture, suggesting that SASP components secreted by these melanocytes mediate adverse paracrine effects [40]. In addition, keratinocytes in the presence of senescent melanocytes have increased the expression of aging markers and reduced proliferation. Interestingly, removing aged melanocytes with the senolytic drug ABT737 caused inhibition of aging and thickening of the epidermis. Similar results were obtained with the MitoQ antioxidant, targeting mitochondria, indicating the critical role of oxidative stress in skin senescence. Senescent melanocytes also contribute to age-related epidermal atrophy, inducing telomere damage and aging in surrounding keratinocytes and fibroblasts [41].

### 2.4. Langerhans Cells

Aging introduces several changes in the skin immune system, including a reduced number of Langerhans cells, decreased antigen-specific immunity, and increased regulatory populations (e.g., regulatory T cells). These alterations result in reduced immunity in the elderly, leading to the increased susceptibility to cancer and infections. In addition, Langerhans cells from older donors have a reduced capacity to migrate to the lymph nodes [42] and express less human b-defensin-3, an antimicrobial peptide [43].

## 3. The Influence of Senescent Cells and SASP on Skin Function

The prolonged presence of senescent cells within tissues and their secretome contribute to aging-related tissue decline and cancerogenesis. However, senescence and SASP constitute a protective mechanism preventing the transformation of damaged cells into tumor cells and play an essential physiological role in wound healing.

### 3.1. Cellular Senescence and Wound Healing

Senescent cells play a complex role during normal wound healing and in chronic wounds. Research conducted by Demaria et al. showed that senescent cells accumulate during wound healing and secrete platelet-derived growth factor AA (PDGF-AA) to induce myofibroblast differentiation and maturation needed for wound closure [44]. Elimination of senescent cells reduces the number of myofibroblasts, delaying wound healing and increasing fibrosis [45]. In contrast, senescent cells in elderly skin prevent wound closure, resulting in chronic wounds. Moreover, in skin exposed to radiation, the accumulation of senescent cells promotes the formation of radiation ulcers, and their elimination (e.g., with dasatinib and quercetin treatment) accelerates the healing process [46].

This phenomenon can be partially explained by the existence of two types of senescent cells. “Short-lived” cells act as positive regulators of wound healing because they promote the formation of granulation tissue, tissue remodeling, and prevent the hyperproliferation of potentially premalignant or malignant cells. Conversely, “long-lived” or chronic tissue senescent cells significantly delay the healing process by creating a tissue environment with chronic inflammation that promotes collagen degradation [26,47,48].

### 3.2. Skin Senescence and Cancerogenesis

Cell senescence prevents uncontrolled cell proliferation, inhibiting tumor formation. SASP production is crucial for recruiting immune cells with anti-tumor activity. However, senescent cells and SASP can also contribute to cancer development [49]. Chronic exposure to SASP can create a tumor-favoring tissue microenvironment that promotes malignant phenotypes in vitro and in vivo [34]. For instance, while several components of the SASP produced by fibroblasts are essential for skin remodeling and repair, some (e.g., IL-6, IL-8, and certain microRNAs) can contribute to cancer cell migration, growth, invasion, angiogenesis, and eventually metastasis [50,51,52]. Interestingly, non-senescent cancer-associated fibroblasts have a secretory pattern resembling SASP, suggesting that targeting SASP can increase the effectiveness of cancer therapy [53].

## 4. Therapeutic Strategies Targeting Skin Senescence

Due to the harmful effects of senescent cells and SASP components on many tissues, strategies aimed at selective induction of senescent cell death or inhibiting SASP without affecting the selective induction of death of surrounding cells are currently being investigated [54]. Removal of senescent cells from aging tissues is considered a promising anti-aging therapy. However, under certain circumstances, such skin cells can also play a positive role [55]. Therefore, SASP modification and maintaining the beneficial features of cell senescence seem to be a more rational therapeutic approach than senescent cell removal.

Complex signaling pathways control SASP production. Nuclear factor κ-light-chain enhancer of activated B cells (NF-κB) is a crucial transcription factor for SASP induction. However, the DNA damage response (DDR), p38 mitogen-activated protein kinase (MAPK), CCAAT/enhancer-binding protein b (C/EBPb), mechanistic target of rapamycin (mTOR), phosphoinositide-3-kinase (PI3K), Janus kinase/signal transducer and activator of transcription (JAK/STAT), protein kinase D1, and several other factors are also involved in regulating SASP production by senescent cells [56].

Different drugs specifically block the signals associated with senescent cell secretion. For example, glucocorticosteroids can reduce SASP secretion and inflammation induced by senescent cells and SASP due to their ability to decrease the transcriptional activity of NF-κB [2]. However, several adverse side effects of glucocorticoid treatment (e.g., skin thinning and impaired wound healing) limit their application as skin senolytics [57]. Other approved SASP regulators are the antidiabetic drug metformin (1,1-dimethyl biguanide) and the antibiotic and immunosuppressant, rapamycin, which both interfere with the NF-κB and mTOR pathways and slow down the aging process [23]. There is growing evidence that flavonoids can prevent skin from aging by targeting cellular pathways crucial for regulating cellular senescence and SASP production.

## 5. Flavonoids as a Senostatic and Senolytic Strategy

Flavonoids are natural substances with variable phenolic structures containing 15 carbon atoms. They consist of two benzene rings connected by a short three-carbon chain. One of the carbons in this chain is connected to carbon in one of the benzene rings, either through an oxygen bridge or directly yielding a third middle ring [58], Figure 1. To date, over 8000 different flavonoids have been identified [59].

Flavonoids are divided into different subtypes: flavones, flavonols, isoflavones, flavanones, anthoxanthins, anthocyanins, and chalcones. They are present in fruits, vegetables, grains, flowers, tea, and wine, and are well-known for their beneficial effects on health. Flavonoids are an indispensable component of various pharmaceutical, medical, and cosmetic applications due to their antioxidative, anti-inflammatory, anti-mutagenic, and anti-carcinogenic properties coupled with their capacity to modulate critical enzyme functions. All these features make flavonoids excellent candidates for anti-aging therapies. 

The enhanced binding of NF-κB to nuclear DNA is one of the hallmarks of aging and is observed in several tissues. NF-κB is a critical transcription factor involved in the production of SASP and the pathogenesis of many age-related disorders, including inflammatory and metabolic diseases [60]. Several flavonoids can disrupt the activation of NF-κB and related pathways, including the kinase 1 signaling pathway associated with the IL-1 receptor (IRAK1)/IκBα and IκBζ, which blocks SASP in vitro [61]. Structural analyses using synthetic flavones revealed that hydroxyl substitutions at C-2′, 3′, 4′, 5’, and 7’ are essential in inhibiting SASP production [62]. Furthermore, flavonoids have a protective effect in animal models of age-related disorders by preventing increased production of IL-1β and tumor necrosis factor (TNF)-α [63].

In this review, we focused on select representatives of flavones, flavonols, isoflavones, and flavanones, whose anti-inflammatory potential in the context of skin cell senescence has been demonstrated in vitro or in vivo (Figure 1). However, it should be mentioned that several other compounds from the group of flavonoids (e.g., curcumin) are being tested for their senolytic and senostatic properties in the context of skin disorders [64].

### 5.1. Flavones

Flavones occur in a wide variety of fruits, vegetables, and cereal grains in the form of glycosides. As with other flavonoid glycosides in foods, flavones must be hydrolyzed to aglycones to be absorbed. They are then metabolized to glucuronidated or sulfated forms before reaching systemic circulation. The main flavones in the diet are apigenin and luteolin; however, some other compounds (e.g., baicalin and wogonin) are also worth mentioning [65].

#### 5.1.1. Apigenin

Apigenin, a flavone present in select fruits, vegetables, and herbs, can induce apoptosis and inhibit proliferation and angiogenesis in several cancer cell lines [66]. The anti-cancer activities of apigenin result from its ability to interact with the PI3K/protein kinase B (ERK)/mTOR, JAK/STAT, NF-κB, MAPK, and Wnt/β-catenin pathways [67]. Interference with mTOR signaling is a dominant mechanism by which apigenin inhibits skin cancer development and progression [68]. Moreover, apigenin has antioxidant and anti-inflammatory properties and can restore the proper function of the skin (e.g., DNA repair and viability of human keratinocytes and dermal fibroblasts) after damage caused by exposure to UVA and UVB radiation [69,70,71]. The molecular mechanisms underlying these phenomena involve the ability of apigenin to inhibit the expression of cyclooxygenase-2 (COX-2) and the NF-κB pathway, which controls the inflammation caused by UVA and UVB radiation [66]. The interaction between apigenin and the NF-κB pathway also seems to be a key mechanism for reducing the secretion of several SASP factors (e.g., IL-6 and IL-8) in human fibroblasts induced to undergo senescence by bleomycin [62]. Moreover, topical administration of apigenin to mice exposed to UVB radiation reduced cutaneous inflammation by inducing thrombospondin 1 (TSP-1) expression and repressing IL-6 and IL-12 levels and inflammatory infiltrates [72].

Aging is associated with increased interferon-γ-inducible protein 10 (IP10) levels that can elicit abnormal immune responses in the elderly [73]. Interestingly, apigenin inhibits the production of IP10, a component of SASP secreted by senescent fibroblasts. IP10 and other chemokines (CXCL9 and CXCL11) promote a Th1 response to cellular damage. Apigenin protects the skin against UVA and UVB radiation-induced destruction of the collagen matrix, which causes loss of elasticity and skin dryness, by decreasing the activity of MMP-1. It also induces collagen type I and III de novo synthesis in dermal fibroblasts in vitro and increases dermal thickness and collagen deposition in the dermis in vivo in mice [74,75]. These anti-aging effects of apigenin were confirmed in clinical trials; its topical application improves markers of skin aging, such as firmness, elasticity, and fine wrinkling and maintains hydration [70,76].

#### 5.1.2. Baicalin

Baicalin is a flavone isolated from the roots of *Scutellaria lateriflora Georgi* (Huang Qin In China) that plays a role in skin protection against UVB-induced photodamage [77]. This function is related to its anti-inflammatory and antioxidative properties through modulating NF-κB, COX-1, and inducible nitric oxide synthase (iNOS) activity [78]. By inhibiting the UV-induced generation of ROS in fibroblasts, baicalin prevents the activation of transcription factors (e.g., activator protein 1, AP-1) responsible for the transcription of MMP-encoding genes and subsequent collagen degradation. The senolytic properties of baicalin are not limited to its effects on SASP. This flavone can also decrease the percentage of β-galactosidase-positive cells and p16, p21, and p53 expression in UVB-treated fibroblast cultures [79]. Moreover, treatment of skin fibroblasts with baicalin reduces the number of DNA double-strand breaks induced by UVB [79]. Anti-mutagenic properties of baicalin were also demonstrated in keratinocytes, where this flavone prevented the formation of oxidative adducts induced by UVC [21]. However, it should be stressed that baicalin does not affect cells that have not been exposed to UV irradiation.

#### 5.1.3. Luteolin

The flavone luteolin is a glycoside found in flowers, herbs, vegetables, and spices. After consumption, it is metabolized to the active aglycone, which has antioxidative properties due to the unique luteolin chemical structure. The C2–C3 double bond donates a hydrogen/electron and stabilizes the radical species and the oxo group at C4 that binds transitional metal ions (e.g., iron and copper) to prevent oxidative damage. By decreasing ROS production, luteolin modulates several cellular pathways, including MAPK and NF-κB, and several downstream genes (e.g., COX-2, IL-6, IL-1β, TNF-α), producing an anti-inflammatory effect [80]. These properties are of particular importance in the context of skin photoaging. Luteolin reduces UV-induced ROS production and subsequent release of pro-inflammatory cytokines (e.g., IL-6 and IL-20) from keratinocytes and MMP-1 from fibroblasts [81,82]. By decreasing ROS production, luteolin prevents increased degradation of hyaluronic acid, which, together with collagen, is the major non-fibrous component of the dermis and epidermis extracellular matrix [83]. Moreover, luteolin alone or in combination with apigenin can directly inhibit UVB-induced MMP-1 production in fibroblasts by inhibiting the Ca^2+^ influx that prevents the phosphorylation of Ca^2+^/calmodulin-dependent MAPKs and binding of the AP-1 transcription factor to the promoter of the MMP-1 gene [84,85].

#### 5.1.4. Wogonin

Wogonin is a flavone extracted from *Scutellaria baicalensis* with proven efficacy as a SASP regulator in cancer [86]. By inactivating the MAPK/AP-1 and NF-κB/IκBα signaling pathways, wogonin downregulates COX-2 and iNOS expression in skin fibroblasts and MMP-1 and IL-6 in UVB-induced keratinocytes [87,88]. Moreover, treatment with wogonin effectively restores procollagen type I levels and increases the expression of cytoprotective antioxidants (e.g., heme oxygenase-1 [HO-1] and NAD(P)H dehydrogenase [quinone] 1 [NQ-O1]) in keratinocytes by activating the tumor growth factor β (TGF-β)/Smad pathway [88]. Wogonin also reduces dermis prostaglandin E2 (PGE2), TNF-α, intercellular adhesion molecule-1 (ICAM1), and IL-1β levels in an animal model of skin inflammation when applied topically [87,89,90].

### 5.2. Flavonols

Flavonols are the most ubiquitous flavonoids in foods, including fruits, vegetables, red wine, and tea, and are represented by quercetin, kaempferol, and fisetin. Like other flavonoids, flavonols accumulate in plant tissue in glycosylated forms linked to mono-, di-, and tri-saccharides. Due to their antioxidative, anti-inflammatory, anti-cancerogenic, and vasodilating properties, flavonols have many benefits to human health, including their effects on senescence [91].

#### 5.2.1. Quercetin

Quercetin is present in red wine, fruits, and vegetables. It can interact with protein kinase C (PKC) δ and Janus kinase 2 (JAK2) to block UV-induced expression of COX-2 and MMP-1 and collagen degradation in human skin and skin fibroblasts [92]. JAK2 kinase is an upstream regulator of STAT3. The STAT3 pathway is involved in stimulating inflammatory responses. In turn, PKCδ is a regulator of the MAPK and Akt signaling pathways and modulates the expression of collagen genes in skin cells [93]. Similar findings came from the study with quercetin surface-functionalized Fe_3_O_4_ nanoparticles (MNPQ). MNPQ-stimulated 5′ AMP-activated protein kinase (AMPK) activity in skin fibroblasts is accompanied by a decrease in the number of stress-induced senescent cells and suppression of senescence-associated secretion of the inflammatory mediators IL-8 and interferon-β [94]. In keratinocytes, quercetin decreases UV-induced activation of NF-κB, resulting in suppressed expression of IL-1β, IL-6, IL-8, and TNF-α. It did not affect UV-mediated activation of ERK, JNK, or p38. Moreover, the induction of AP-1 target genes (e.g., MMP-1 and MMP-3) is not suppressed by quercetin [95]. Apart from being senostatic, quercetin also has senolytic properties. The combination of dasatinib and quercetin effectively eliminates senescent fibroblasts in vitro and reduces senescence of primary mouse embryonic fibroblasts (MEFs) in vivo in chronologically aged or radiation-exposed mice and also progeroid mice model [8].

#### 5.2.2. Kaempferol

The flavonol kaempferol is found in many edible or traditional medicine plants and has antioxidant and anti-inflammatory properties by inhibiting the iNOS, COX-2, and NF-κB pathways [96]. Administration of kaempferol to aged (24-week-old) rats reduces the accumulation of advanced glycation endproducts (AGE) in different organs and decreases the expression of AGE receptor (RAGE) and AGE-induced reactive species (RS). Because RS are potent activators of NF-κB, both kaempferol-treated fibroblasts and animals have lower expression of MMP-9, adhesion molecules (e.g., ICAM-1), and several pro-inflammatory genes. Accordingly, in bleomycin-induced senescent fibroblasts and aged rats, kaempferol inhibits the induction of a subset of SASP mRNA and the activation of the NF-κB pathway [62].

#### 5.2.3. Fisetin

Fisetin is a flavonol with a chemical structure similar to quercetin. It is present in many fruits and vegetables (e.g., apples, persimmon, grapes, onions, and cucumbers) at relatively low concentrations and at high concentrations in strawberries. Fisetin has demonstrated potent senolytic and senostatic properties in vitro and in vivo. Administration of fisetin to progeroid and old wild-type mice reduces senescence markers (i.e., p16 and p21), modifies the SASP composition in multiple tissues, and restores tissue homeostasis by inhibiting the PI3K/AKT/mTOR and NF-κB pathways and antioxidative activity [97].

In the context of skin aging, fisetin can inhibit TNF-α-induced inflammation and hydrogen peroxide-induced oxidative damage in human keratinocytes [98]. It can also decrease UVB-induced damage by inhibiting ROS generation and the MAPK/AP-1/MMP signaling pathway and decreasing collagen degradation and the inflammatory response in human skin fibroblasts [99]. When applied topically to hairless mice, fisetin inhibits iNOS, MMP-1, MMP-2, and COX-2 and increases the skin expression of filaggrin and aquaporins, protecting the animals from photo-inflammation and skin-drying [100]. Clinical trials are currently underway to evaluate the benefits of fisetin treatment on several aspects of aging [101].

### 5.3. Isoflavones

Isoflavones are non-active hydrophilic glycosides (e.g., daidzin and genistein in soybean) or methylated lipophilic derivatives (e.g., formononetin and biochanin A in red clover) in the plants from the *Leguminosae* family that are hydrolyzed by β-glucosidases in the gastrointestinal tract. These bioactive aglycones (e.g., daidzein and genistein formed from daidzin and genistin, respectively) are absorbed across the intestinal epithelium and metabolized to β-glucuronides and sulfate esters in the intestinal mucosa cells. These metabolites are subsequently excreted into plasma and bile [102].

The pleiotropic effects of isoflavones depend on their ability to interact with several nuclear receptors, including estrogen receptors (ER) α and β; peroxisome proliferator-activated receptors (PPARs) α, δ and γ; retinoid acid receptor (RAR); and aryl hydrocarbon receptor (AhR). However, the isoflavones also act by nuclear receptor-independent mechanisms, including inhibition of protein tyrosine kinases (e.g., ERK1/2, crucial for regulating cell proliferation and differentiation), reduction of ROS levels, induction of antioxidant enzymes, and inhibition of COX-1 and NF-κB activity and thromboxane A_2_ (TXA_2_) synthesis. All these functions contribute to the anti-inflammatory properties of the isoflavones [60].

#### Daidzein and Genistein

Daidzein alone or in combination with genistein inhibits UV-induced MMP-1 and MMP-2 expression and collagen degradation in human skin fibroblasts in vitro and in hairless mice in vivo [103]. UV radiation can disrupt the skin collagen matrix by inhibiting the TGF-β pathway [94]. Daidzein increases TGF-β expression and activates its receptors (signal transducer and activator of transcription 2/3—Smad2/3) in skin fibroblasts. Importantly, daidzein does not affect skin cell viability [104]. Moreover, through its interaction with RAR in human keratinocytes, daidzein can inhibit the expression of MMP-9, a metalloproteinase involved in the development of chronic ulcers in diabetic patients [105,106].

Genistein prevents UV-dependent COX-2 expression in human keratinocytes in vitro and the release of pro-inflammatory mediators [107]. Moreover, topical genistein or its metabolite equol protects against UVB-induced oxidative DNA damage (DNA pyrimidine dimer formation) and ROS production in the skin of hairless mice [108]. Like daidzein, genistein increases the thickness of skin collagen fibers by inducing TGF-β expression and increasing tissue inhibitor of metalloproteinase (TIMP) protein levels [109]. Both genistein and daidzein have significant anti-inflammatory effects and promote genomic and mitochondrial DNA repair in human skin fibroblasts exposed to UVB radiation (REF). They also work synergistically to produce a photoprotective effect [110,111]. Moreover, daidzein and genistein stimulate the production of hyaluronic acid in transformed human keratinocyte culture and hairless mouse skin [112].

There are studies suggesting that the administration of isoflavones can reverse the symptoms of skin aging in humans. For instance, 12-week systemic treatment with 40 mg of soy isoflavone aglycones improved fine wrinkles and skin elasticity in middle-aged Japanese women [113]. However, 24-week topical genistein administration had no superiority over estradiol and was less effective than this hormone in improving epidermal thickness, the number of dermal papillae, fibroblasts, and vessels in post-menopausal women [114].

### 5.4. Flavanones

Flavanones are found chiefly in citrus fruits; the most abundant flavanone is naringenin present in grapefruits, lemons, tangerines, and oranges. Naringenin has many pharmacological properties, including anti-atherogenic, anti-cancer, antioxidant, and anti-inflammatory. In the context of skin aging, naringenin can protect human keratinocytes against UVB-induced carcinogenesis and aging in vitro and UVB-generated oxidative stress and inflammation in vivo [115,116]. Topical naringenin protects hairless mice from UVB-induced skin damage by inhibiting the production of SASP components (TNF-α, IL-1β, IL-6, and IL-10) and lipid hydroperoxides, while maintaining the expression of antioxidant genes, including glutathione peroxidase 1, glutathione reductase, and the nuclear factor erythroid 2-related factor 2 (Nrf2) transcription factor [117]. These effects are partly due to the ability of naringenin to decrease NF-кB, MMP-1, and MMP-3 levels [118].

The mechanisms of the senostatic and senolytic actions of the different flavonoid subtypes in the context of skin aging are summarized in Table 1.

## 6. Summary and Conclusions

Targeting senescent cells has become an alternative therapy for treating various age-related conditions and diseases. This targeting can be achieved on two levels: specific elimination of senescent cells and inhibition of their secretory phenotype. Because senescent cells play a significant role in skin physiology and pathophysiology, their elimination may have unpredictable adverse effects. Therefore, modulation of SASP may be a safer strategy to counteract the senescence of skin cells. In vitro and in vivo studies suggest that the administration of flavonoids both topically and systemically has many benefits in this regard. However, due to the heterogeneity of study protocols, these pre-clinical findings cannot be translated directly into clinical practice. Therefore, we still lack convincing clinical studies to confirm the effectiveness and safety of flavonoids in treating age-related skin changes and lesions. Additional research is needed to optimize the appropriate treatment and assess the potential adverse effects of flavonoid applications. Clinical trials must be supported by solid pre-clinical results obtained in appropriate cellular and animal models. It is also necessary to develop a treatment scheme and appropriate cell markers to assess the effectiveness of the therapy. Moreover, research protocols should be unified so that the results obtained with different research models are comparable and translatable to clinical practice.

Taking into account the potential beneficial effect of flavonoids on skin aging, a diet rich in vegetables, fruits, and cereals, which are a natural source of these compounds, should be recommended in general anti-aging management. Importantly, natural products constitute a mixture of various flavonoids that can act comprehensively and synergistically and, therefore, are more effective than compounds evaluated in experimental settings. Furthermore, since flavonoids in natural products are present in mild/moderate concentrations, they can be safely administered without the risk of overdosage. Moreover, pre-clinical trials demonstrated a wide safe therapeutic range of flavonoids. Therefore, nutraceuticals and dietary supplements containing both natural flavonoids as well as semi-synthetic and synthetic compounds with a variety of substituents and proven activity may be considered as a rational method of preventing skin aging.

## Figures and Tables

**Figure 1 ijms-22-06814-f001:**
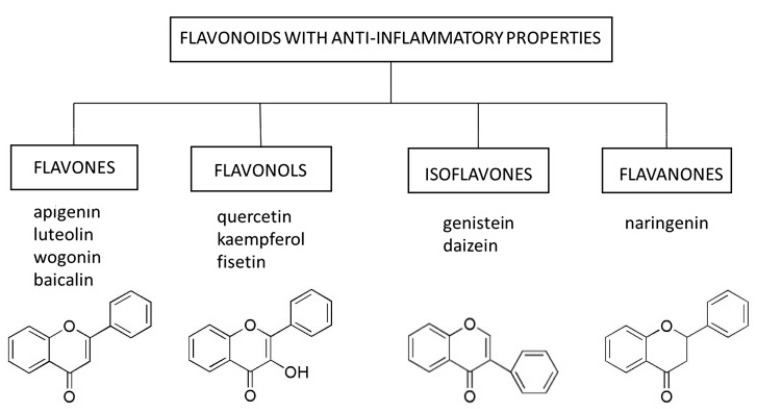
Select flavonoids with anti-inflammatory activity.

**Table 1 ijms-22-06814-t001:** Mechanisms of senolytic and senostatic actions of flavonoids in the context of skin aging.

Flavonoid	Route of Administration	Disease Entity	Research Model	Mechanism	Conclusion	Reference
**Flavones**						
Apigenin	in vitro	UVA & UVB-induced skin ageing	human dermal fibroblasts	↓ NF-κB pathway ↓ MAPK ↓ MMP-1 ↓ ROS	↑ viability ↑ collagen synthesis ↑ DNA repair	[70] [74] [71]
	in vitro	UVB-induced skin damage	human keratinocytes	↑ TSP-1	↓ UVB-induced carcinogenesis	[69]
	topical (25 μmol/μL)	UVB-induced acute skin damage	mice	↑ TSP-1 ↓ IL-6, -12 ↓ inflammatory infiltrates	restoration of skin damage caused by UVB radiation	[72]
	topical (1.5–3 mg/cm^2^)	UVA-induced skin ageing	mice	↓ NF-κB pathway ↓ MAPK ↓ MMP-1 ↓ ROS	↑ dermal thickness ↑ collagen deposition	[74]
	in vitro	bleomycin-induced cellular senescence	human dermal fibroblasts	↓ NF-κB pathway ↓ IL-1α, -1β, -6, -8 ↓ GM-CSF, CXCL1 ↓ MCP-2, MMP-3	↓ SASP secretion	[62]
	in vitro	ionizing radiation-induced cellular senescence	human dermal fibroblasts	↓ NF-κB pathway ↓ MAPK ↓ IP10	↓ SASP secretion	[73]
	topical (2 g of 1% cream for 4 weeks)	UVA-induced skin ageing	healthy individuals	↓ MMP-1	↑ dermal density and elasticity ↓ fine wrinkle length	[70]
Baicalin	in vitro	UVB-induced skin ageing	human dermal fibroblasts human skin samples	↓ MMP-1, MMP-3 ↓ ROS ↓ p16, p21,p53	↑ collagen synthesis ↓ DNA damage ↓ apoptosis ↑ viability	[79]
	in vitro	UVC-induced cytotoxicity	human keratinocytes	↓ ROS	↓ DNA damage	[21]
	topical (0.5–1 mg/cm^2^)	UVB-induced skin damage	Balb/C mice	↓ p53 in epidermis	↓ DNA damage ↓ apoptosis	[77]
Luteolin	in vitro	UVB-induced skin damage	human keratinocytes	↓ p38/MAPK ↓ ROS ↓ COX-2	↓ DNA damage	[82]
	topical (0.25–4 mg/mL)		human skin explants			
	in vitro	UVA-induced skin ageing	human dermal fibroblasts human keratinocytes	↓ p38/MAPK ↓ ROS ↓ MMP-1 ↓ IL-6, -20	↓ SASP secretion ↓ collagen degradation ↓ hylauronic acid degradation	[81]
	topical (8 mg/mL)		human skin explants			
	in vitro	UVB-induced skin ageing	human dermal fibroblasts	↓ MAPK/AP-1 ↓ NF-κB pathway ↓ MMP-1	↓ SASP secretion ↓ collagen degradation	[85]
Apigenin & Luteolin		UVA-induced skin damage	human keratinocytes	↓ MAPK/AP-1 ↓ MMP-1	↓ SASP secretion ↓ collagen degradation	[84]
Wogonin	in vitro		mouse dermal fibroblasts	↓ COX-2	↓ SASP secretion	[87]
	in vitro	UVB-induced skin damage	human keratinocytes	↓ MAPK/AP-1 ↓ NF-κB pathway ↓ IL-6 ↓ MMP-1 ↓ TGF-β ↓ Nrf2	↓ SASP secretion ↑ collagen synthesis ↑ antioxidants	[88]
**Flavonols**						
Quercetin	in vitro	UV-induced skin ageing	human dermal fibroblasts human skin explants	↓ MAPK/AP-1 ↓ NF-κB pathway ↓ JAK2/STAT3 ↓ COX-2 ↓ MMP-1	↓ SASP secretion ↓ collagen degradation	[92]
	in vitro	hydrogen peroxide-induced skin ageing	human dermal fibroblasts	↑ AMPK ↓ IL-8, IFN-β	↓ SASP secretion ↓ senescent cells number	[94]
	in vitro	UV-induced skin damage	human keratinocytes	↓ NF-κB ↓ IL-1β, -6, -8,TNF-α =MMP-1, -3	↓ SASP secretion	[95]
Kaempferol	in vitro	bleomycin-induced senescence	fibroblasts	↓ NF-κB pathway	↓ SASP secretion	[62]
Fisetin	in vitro	hydrogen peroxide-induced skin damage	human keratinocytes	↓ ROS ↓ NF-κB pathway ↓ iNOS ↓ COX-2 ↓ IL-1β, -6, TNF-α	↑ viability ↓ SASP secretion	[98]
	in vitro	UVB-induced skin damage	human dermal fibroblasts	↓ MAPK/AP-1/MMP ↓ ROS	↓ SASP secretion ↓ collagen degradation	[99]
	topical (50–200 μmol/daily)	photoinflammation	hairless mice	↓ iNOS ↓ MMP-1, -2 ↓ COX-2 ↑ filaggrin ↑ aquaporins	↓ SASP secretion ↓ collagen degradation ↓ photo inflammation ↓ skin-drying	[100]
**Isoflavones**						
Daidzein & Genistein	in vitro	UVB-induced skin damage	human keratinocytes	↓ MMP-1, -2	↓ collagen degradation	[103]
	systematically (500 mg of soy extract/kg/day)		hairless mice		↓ wrinkle length	[103]
	in vitro		human keratinocytes	↑ hyaluronic acid		[112]
	topicaly (0.1 mL of 10 μmol equol solution)		hairless mice		↓ fine wrinkle	[108]
	in vitro	UVB-induced skin damage	human dermal fibroblasts	↓ COX-2 ↓ Gadd45	↑ genomic and mitochondrial DNA repair	[110] [111]
	systemically (40 mg of soy isoflavone aglycone per day)	estrogen deficiency	middle-aged women		↓ fine wrinkles ↑ skin elasticity	[113]
Daidzein	in vitro	UV-induced skin damage	human dermal fibroblasts	↑ TGF-β/Smad2/3	↓ collagen degradation	[104]
	in vitro	particulate matter-exposure	human keratinocytes	↓ MAPK ↓ COX-2 ↓ MMP-9	↓ SASP	[106]
Genistein	in vitro	UV-induced skin damage	human keratinocytes	↓ COX-2	↓ SASP	[107]
	topical (0.1 mL of 10 μmol equol solution)	UVB-induced skin damage	hairless mice	↓ DNA pyrimidine dimer formation ↓ ROS	↓ DNA damage	[108]
	systematically (1 mg/kg sc)	estrogen deficiency	ovariectomized rats	↑ TGF-β/Smad2/3 ↑TIMP ↓ TGF-β, MMP-2,-9	↓ collagen degradation	[109]
**Flavanones**						
naringenin	in vitro	UVB-induced apoptosis	human keratinocytes	↑ caspase cascade pathway	↓ apoptosis	[115]
	intraperitoneal (10–100 mg/kg)	UVB-induced inflammation	hairless mice	↓ ROS ↓ MMP-9, ↓ TNF-α, IFN-γ, ↓ IL-1β, -4,-5,-6,-12, -13, -17, -22, -23	↓ SASP ↓ inflammatory infiltrations	[116]
	topical (0.5% solution)	UVB-induced skin damage	hairless mice	↓ ROS ↓ TNF-α, IL-1β, -6, -10	↓ SASP	[117]
	in vitro	LPS-induced skin damage	human dermal fibroblasts	↓ NF-κB pathway ↓ MMP-1,-3	↓ SASP ↓collagen degradation	[118]

AP-1—activator protein 1 transcription factor; COX—cyclooxygenase; CXCL—chemokine (C–X–C motif) ligand 1; DMSO—dimethyl sulfoxide; Gadd45—growth arrest and DNA damage; GM–CSF—granulocyte–macrophage colony–stimulating factor; IFN—interferon; IL—interleukin; iNOS—inducible nitric oxide synthase; IP10—interferon-γ-inducible protein 10; JAK2—Janus kinase 2; LPS—lipopolysacharide; MAPK—mitogen-activated protein kinase; MCP—methyl chemotaxis protein; MMP—matrix metalloproteinase; NF-κB—nuclear factor κ-light-chain enhancer of activated B cells; Nrf2—nuclear factor erythroid 2-related factor 2; ROS—reactive oxygen species; SASP—senescence-associated secretory phenotype; Smad—SMAD protein; STAT3—signal transducer and activator of transcription; TGF-β—tumor growth factor β; TIMP—tissue inhibitor of metalloproteinase proteins; TNF-α—tumor necrosis factor α; TSP-1—thrombospondin 1; UV—ultraviolet; ↓—decrease; ↑—increase.

## Data Availability

Not applicable.

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
