# Peer review of "Flavonoids in Skin Senescence Prevention and Treatment"

_ijms, 2021, doi:10.3390/ijms22136814_

Round 1

Reviewer 1 Report

The article addresses important issues related to skin aging, wound healing.

The work needs minor proofreading (some sentences are too long, or need to be checked for grammar).

In recent years, there has been a lot of information on curcumin and its anti-inflammatory properties in the context of wound healing. It may be worth adding such information. But this I leave to the discretion of the Authors.   

Please review your manuscript once for typos, commas and sentence length. 

Author Response

Reviewer 1

  • The article addresses critical issues related to skin aging, wound healing. The work needs minor proofreading (some sentences are too long, or need to be checked for grammar).

We thank the Reviewer for the generally positive assessment of our manuscript. Since none of the authors is a native speaker, we know that the manuscript requires English editing. That is why we took advantage of the professional service, and the manuscript has been copy-edited by native English speakers with related biomedical backgrounds in Biomed Proofreading® LCC (please see the attached certificate).

  • In recent years, there has been much information on curcumin and its anti-inflammatory properties in the context of wound healing. It may be worth adding such information. But this I leave to the discretion of the Authors.

We thank the Reviewer for this valuable suggestion. We agree that several reports have been published regarding the anti-inflammatory effects of curcumin in the context of wound healing in recent years. However, in our review, we decided to focus primarily on compounds that could be used to prevent skin aging, and wound healing is a side topic raised only to demonstrate the physiological functions of senescent cells. However, to make this point more precise, we added relevant information in the revised version of the manuscript.

“In this review, we focused on select representatives of flavones, flavonols, isoflavones, and flavanones, whose anti-inflammatory potential in the context of skin cell senescence has been demonstrated in vitro or in vivo (Figure 1). However, it should be mentioned that several other compounds from the group of flavonoids (e.g., curcumin) are being tested for their senolytic and senostatic properties in the context of skin disorders [64].” (Page 6, lines 282-287)

[64] Vollono, L.; Falconi, M.; Gaziano, R.; Iacovelli, F.; Dika, E.; Terracciano, C.; Bianchi, L.; Campione, E. Potential of Curcumin in Skin Disorders. Nutrients 2019, 11, 2169. https://doi.org/10.3390/nu11092169.

  • Please review your manuscript once for typos, commas and sentence length. 

We have made every effort to improve the manuscript in editing (including English editing) and apologize for the previous carelessness.

Reviewer 2 Report

The article entitled „Flavonoids in skin senescence prevention and treatment” submitted for evaluation reviews flavonoids application as preventive and therapeutic strategy for skin ageing physiologically or ageing skin exposed to UV radiation, X-radiation or cigarette smoke, including molecular mechanism of their action observed in in vitro and in vivo tests.

The manuscript is well-organized, clear and complete review, with easy to follow, logical flow, equipped with a wild selection of up-to-date references (108 items in the field).

Minor remarks are as follows:

  • In subsection (2.3) "Melanocytes" it is worth mentioning that melanocytes produce a pigment melanin that protects the skin exposed to UV radiation (from natural sun and the tanning lamps and beds); when UV burning and radiation damage (mutations) occur in the melanocyte uncontrolled cellular growth leads to dangerous skin cancer melanoma that in 70-80% arise in normally looking skin, and elder individuals are of higher risk for developing melanoma; overactive melanocytes are also responsible for excessive pigmentation characteristic for old skin - age spots (sunspots, liver spots) on skin that had the most sun exposure over the years
  • In section 5. "Flavonoids as a senostatic and senolytic strategy" the overall molecular (C6-C3-C6) and structural formulas of flavonoids could be provided; optionally also example of structural formulas may be provided for the discussed flavonoids, e.g. apigenin, baicalin, quercetin
  • line 307: luteoin – should be luteolin
  • line 384: Leguminosae – italics please
  • Table 1: in leftmost column titled "Route of administration" it is worth to add an information “in vitro” for in vitro tests (on human fibroblasts, human keratinocytes et cetera); in addition to add information about safety of flavonoids administration in animal studies (if such information is available in original articles); Quercetin column: IL-1b – should be IL-b (beta); =MMP-1,-3 =?
  • In section 6. "Summary and conclusions" it is worth adding information about necessity vegetable and fruit rich diet which are natural sources of a whole range (mixture) of flavonoids in mild/moderate concentrations, working comprehensively and synergistically, and therefore, being effective and safe; it is worth to add information that as nutraceuticals/diet supplements both natural flavonoids (isolated from fruits, vegetables, cereals), as well as semi-synthetic and synthetic compounds with variety substituents and proven activity may be used

Author Response

Reviewer 2

The article entitled "Flavonoids in skin senescence prevention and treatment" submitted for evaluation reviews flavonoids application as preventive and therapeutic strategy for skin ageing physiologically or ageing skin exposed to UV radiation, X-radiation or cigarette smoke, including molecular mechanism of their action observed in in vitro and in vivo tests. The manuscript is well-organized, clear and complete review, with easy to follow, logical flow, equipped with a wild selection of up-to-date references (108 items in the field).”

We thank the Reviewer for the positive reception of our manuscript and for drawing attention to some weak points we tried to improve.

  • In subsection (2.3)"Melanocytes," it is worth mentioning that melanocytes produce a pigment melanin that protects the skin exposed to UV radiation (from natural sun and the tanning lamps and beds); when UV burning and radiation damage (mutations) occur in the melanocyte uncontrolled cellular growth leads to dangerous skin cancer melanoma that in 70-80% arise in normally looking skin, and elder individuals are of higher risk for developing melanoma; overactive melanocytes are also responsible for excessive pigmentation characteristic for old skin - age spots (sunspots, liver spots) on skin that had the most sun exposure over the years.

Following the Reviewer's suggestion, we extended the subsection regarding melanocytes taking into account the role of melanin in protecting the skin against carcinogenesis:

“Melanocytes contain specialized lysosome-lineage organelles called melanosomes dedicated to synthesis and storage of melanin, a photoprotective pigment that protects skin from UVB, UVA, and visible blue light. Melanin-containing melanosomes can be transferred from melanocytes to the surrounding keratinocytes that together constitute a melano-epidermal unit. Melanin acts as a redox UV-absorbing agent and, in this way, directly prevents the DNA of epidermal cells from photodamage. However, melanin contributes to DNA protection also indirectly by scavenging by scavenging reactive oxygen species (ROS) formed during the UV-inducing oxidative stress in the skin [38]. Aging is associated with several changes in the skin pigmentary system that can be accelerated by exposition to UV radiation, leading to structural changes in melanocytes and their hyperactivity. Ectopic up-regulation of melanocytes contributes to the formation of senile lentigines/lentigo and other age-related hyperpigmentation disorders and may result in the development of melanoma – the most lethal of all types of skin cancers; which incidence is growing with age [39].” (Page 4, lines 166-179)

We also added two additional items to the reference list:

[38] Solano, F. Photoprotection and skin pigmentation: Melanin-related molecules and some other new agents obtained from natural sources. Molecules 2020, 25, 1–18, doi:10.3390/molecules25071537.

[39] Yamaguchi, Y.; Hearing, V.J. Melanocytes and their diseases. Cold Spring Harb. Perspect. Med. 2014, 4, 1–18, doi:10.1101/cshperspect.a017046.

  • In section 5. "Flavonoidsas a senostatic and senolytic strategy" the overall molecular (C6-C3-C6) and structural formulas of flavonoids could be provided; optionally also example of structural formulas may be provided for the discussed flavonoids, e.g. apigenin, baicalin, quercetin.

Following the Reviewer's suggestion, in the revised version of the manuscript, we provided structural formulas of flavonoids groups discussed in the text. To facilitate the perception of the text, the formulas of each group of flavonoids were attached to Figure 1.

  • “line 307: luteoin – should be luteolin & line 384:Leguminosae – italics please”

We apologize for the typos in the manuscript. The suggested changes were introduced in the text.

  • Table 1:in leftmost column titled "Route of administration" it is worth to add an information “in vitro” for in vitro tests (on human fibroblasts, human keratinocytes et cetera); in addition to add information about safety of flavonoids administration in animal studies (if such information is available in original articles);

We thank the Reviewer for these valuable suggestions that would make Table 1 more informative. We added the phrase "in vitro” to the column “Route of administration” wherever in vitro tests were mentioned. In the case of animal and human studies, we added information regarding the concentration range/amount of a compound applied in a particular experiment. It is worth noting that the compounds were well tolerated and proved to be safe in the various concentration ranges used in the experiments.

Quercetin column: IL-1b – should be IL-b (beta); =MMP-1,-3 =?

In addition, the typo in the interleukin 1β name was corrected.

  • In section 6. "Summary and conclusions" it is worth adding information about necessity vegetable and fruit rich diet which are natural sources of a whole range (mixture) of flavonoids in mild/moderate concentrations, working comprehensively and synergistically, and therefore, being effective and safe; it is worth to add information that as nutraceuticals/diet supplements both natural flavonoids (isolated from fruits, vegetables, cereals), as well as semi-synthetic and synthetic compounds with variety substituents and proven activity may be used.

Following the Reviewer's suggestion, we have added a new paragraph regarding the utility of nutraceuticals/diet supplements containing natural and semi-synthetic flavonoids to prevent skin senescence to the "Summary and conclusions" section.

"Taking into account the potential beneficial effect of flavonoids on skin aging, a diet rich in vegetables, fruits, and cereals, which are a natural source of these compounds, should be recommended in general anti-aging management. Importantly, natural products constitute a mixture of various flavonoids that can act comprehensively and synergistically and, therefore, being more effective than compounds evaluated in experimental settings. Furthermore, since flavonoids in natural products are present in mild/moderate concentrations, they can be safely administered without the risk of overdosage. Moreover, pre-clinical trials demonstrated a wide safe therapeutic range of flavonoids. Therefore, nutraceuticals and dietary supplements containing both natural flavonoids as well as semi-synthetic and synthetic compounds with variety substituents and proven activity may be considered as a rational method of preventing skin aging." (Page 18, lines 517-527)

Reviewer 3 Report

This is a thorough and comprehensive review on flavonoids and aging. I read it with interest.

I do have one comment/concern, regarding the accuracy of correctly citing references.

Point 2.1: Aging of keratinocytes: The paper you cite mentions the photoaging effect of the whole UV spectrum. UVA and UVB are widespread, but UVC has an even higher capacity for mutagenic effects and aging. To be completely correct you shouldn't specify (only mention) UVB. Either mention all or just call the rays UV radiation. There are also several studies that register the effect of infrared radiation on photoaging, alone, and as a factor accelerating UV damage (via MMP). Perhaps this should be included for a completely exhaustive description of radiation and aging.

Likewise, on aging of fibroblasts you write that UVB is the chief stimulus for fibroblast senescence in vitro. Neither of the sited papers (ref. 30 and 31) mention that UVB is the main in vitro, on the contrary, many papers on in vitro stimulating of ageing use UVA+UVB (e.g. in the paper on aging of melanocytes), also many mention X-rays.

Point 3: Senescent cells and SAPS: The claims on the importance of UVB alone are not completely correct and the references sited to support some statements on UVB and ageing actually do not support them. This sets in doubt the correctness of all other references used to back up your statements.

Furthermore, when describing the properties of the different flavinoids you describe many to be favourable in both UVA and UVB damage. It is therefore necessary to be sure UVA ageing mechanisms also are mentioned in the pathophysiology of aging.

Table 1. You refer to reference 62. In the column ‘Disease entity’ you write: UVB induced damage. In ‘Conclusion’ column: Restoration of damage caused by UVA. Obviously, this is incorrect and the paper does not conclude with this.

Please make sure that you site all papers correctly and that the reference actually mentions what you claim that it does. A thorough read-through for checking the validity of all the cited references seems necessary before accepting the paper. I noticed the wrongly cited papers on UVB, but all other entities and conclusions also need to be checked.

Author Response

Reviewer 3

This is a thorough and comprehensive review on flavonoids and aging. I read it with interest. I do have one comment/concern, regarding the accuracy of correctly citing references.

We thank the Reviewer for the positive reception of our manuscript and for drawing attention to some weak points we tried to improve. The main concern of the Reviewer was the accuracy of the selected citations. Indeed, the selection of the cited works was not optimal, and the list of references needed to be extended with additional items, as suggested by the Reviewer. Therefore we introduced several changes to the main text and the Reference list.

  • Point 2.1: Aging of keratinocytes: The paper you cite mentions the photoaging effect of the whole UV spectrum. UVA and UVB are widespread, but UVC has an even higher capacity for mutagenic effects and aging. To be completely correct you shouldn't specify (only mention) UVB. Either mention all or just call the rays UV radiation. There are also several studies that register the effect of infrared radiation on photoaging, alone, and as a factor accelerating UV damage (via MMP). Perhaps this should be included for a completely exhaustive description of radiation and aging.

We are grateful for this valuable suggestion. To better describe the influence of UV and infrared radiation on skin senescence, we added a new paragraph dedicated to this subject:

“UV radiation plays a central role in skin senescence and skin cancer development. UV radiation is composed of three main components based on photon wavelength: UVA having the longest wavelengths (315–400 nm), UVB being mid-range (290–320 nm), and UVC being the shortest wavelengths (100–280 nm). All UV types can act as environmental mutagens leading to direct and indirect (via increased production of oxidative free radicals) DNA damage, and each can result in mutagenesis in skin cells. UVA radiation is the most prevalent component of solar UV radiation. It penetrates deeper than UVB (that has a major action on the epidermis) into the skin and induces profound alterations of the dermal connective tissue [18, 19]. In vitro studies also show that UVC has a deteriorating effect on genome stability, contributing to the aging of fibroblasts and keratinocytes [20, 21]. However, considering that most of this radiation is absorbed by an ozone layer, its clinical relevance is less pronounced. To give the complete picture, it is also important to mention the effects of infrared radiation (IR) on skin aging. Recent studies indicate that IR and heat may induce premature skin aging by stimulation of matrix metalloproteinases (MMP) expression and modulation of elastin and fibrillin synthesis. Moreover, heat stimulates in human skin formation of new vessels, recruitment of inflammatory cells, and causes oxidative DNA damage [22].” (Page 2-3, lines 80-96)

We have also added information about the protective effect of baicalin against the formation of UVC-induced cytotoxicity:

“Antimutagenic properties of baicalin were also demonstrated in keratinocytes, where this flavone prevented the formation of oxidative adducts induced by UVC [21].” (Page 8, lines 345-347 & Table 1)

 With proper references:

[18] Battie, C.; Jitsukawa, S.; Bernerd, F.; Del Bino, S.; Marionnet, C.; Verschoore, M. New insights in photoaging, UVA induced damage and skin types. Exp. Dermatol. 2014, 23, 7–12, doi:10.1111/exd.12388.

[19] Amaro-Ortiz, A.; Yan, B.; D’Orazio, J.A. Ultraviolet radiation, aging and the skin: Prevention of damage by topical cAMP manipulation. Molecules 2014, 19, 6202–6219, doi:10.3390/molecules19056202.

[20] Begović, L.; Antunovic, M.; Matic, I.; Furcic, I.; Baricevic, A.; Vojvoda Parcina, V.; Peharec Štefanić, P.; Nagy, B.; Marijanovic, I. Effect of UVC radiation on mouse fibroblasts deficient for FAS-associated protein with death domain. Int. J. Radiat. Biol. 2016, 92, 475–482, doi:10.1080/09553002.2016.1186298.

[21] Wang, S.C.; Chen, S.F.; Lee, Y.M.; Chuang, C.L.; Bau, D.T.; Lin, S.S. Baicalin scavenges reactive oxygen species and protects human keratinocytes against UVC-induced cytotoxicity. In Vivo (Brooklyn). 2013, 27, 707–714.

[22] Cho, S.; Shin, M.H.; Kim, Y.K.; Seo, J.E.; Lee, Y.M.; Park, C.H.; Chung, J.H. Effects of infrared radiation and heat on human skin aging in vivo. J. Investig. Dermatology Symp. Proc. 2009, 14, 15–19, doi:10.1038/jidsymp.2009.7.

  • Likewise, on aging of fibroblasts you write that UVB is the chief stimulus for fibroblast senescence in vitro. Neither of the sited papers (ref. 30 and 31) mention that UVB is the main in vitro, on the contrary, many papers on in vitro stimulating of ageing use UVA+UVB (e.g. in the paper on aging of melanocytes), also many mention X-rays.

We do apologize for this obvious mistake we tried to correct in the revised version of the manuscript – see the previous section and:

“In contrary to keratinocytes, UVA radiation due to its deeper penetration is the chief stimulus inducing fibroblast senescence in vivo [18, 19], while all types of UV radiation and X-rays have been shown to stimulate fibroblasts senescence in vitro [36, 37].” (Page 4, line 159-162)

With proper references:

[18] Battie, C.; Jitsukawa, S.; Bernerd, F.; Del Bino, S.; Marionnet, C.; Verschoore, M. New insights in photoaging, UVA induced damage and skin types. Exp. Dermatol. 2014, 23, 7–12, doi:10.1111/exd.12388.

[19] Amaro-Ortiz, A.; Yan, B.; D’Orazio, J.A. Ultraviolet radiation, aging and the skin: Prevention of damage by topical cAMP manipulation. Molecules 2014, 19, 6202–6219, doi:10.3390/molecules19056202.

[36] Dong, K.; Goyarts, E.; Rella, A.; Pelle, E.; Wong, Y.H.; Pernodet, N. Age associated decrease of MT-1 melatonin receptor in human dermal skin fibroblasts impairs protection against UV-induced DNA damage. Int. J. Mol. Sci. 2020, 21, 1–13, doi:10.3390/ijms21010326.

[37] Kalfalah, F.; Seggewiß, S.; Walter, R.; Tigges, J.; Moreno-Villanueva, M.; Bürkle, A.; Ohse, S.; Busch, H.; Boerries, M.; Hildebrandt, B.; et al. Structural chromosome abnormalities, increased DNA strand breaks and DNA strand break repair deficiency in dermal fibroblasts from old female human donors. Aging (Albany. NY). 2015, 7, 110–122, doi:10.18632/aging.100723.

  • Point 3: Senescent cells and SAPS: The claims on the importance of UVB alone are not completely correct and the references sited to support some statements on UVB and ageing actually do not support them. This sets in doubt the correctness of all other references used to back up your statements.

We are thankful for this comment. It was not our intention to limit discussion on the influence of radiation on SASP to UVB only. Thus, following the Reviewer’s suggestion, we have added a new paragraph regarding the role of both UVA and UVB in SASP modulation (with proper references). Moreover, as we state below, we made every effort to ensure that the citations in the revised version of the manuscript were adequately matched to the manuscript's content.

“Skin irradiation plays a central role in modulation of SASP, too. While most of UVC is blocked by the ozone layer, the UVA and UVB contribute to skin senescence and inflammation by activating SASP genes like IL-1, IL-6, and MMPs [24]. In turn, both UVA and UVB can downregulate tumor growth factor (TGF)-β resulting in reduced collagen type I synthesis, leading to dermal thinning and wrinkle formation [25]." (Page 3, lines 103-108)

[24] Ghosh, K.; Capell, B.C. The Senescence-Associated Secretory Phenotype: Critical Effector in Skin Cancer and Aging. J. Invest. Dermatol. 2016, 136, 2133–2139, doi:10.1016/j.jid.2016.06.621.

[25] Lee, Y.I.; Choi, S.; Roh, W.S.; Lee, J.H.; Kim, T.G. Cellular senescence and inflammaging in the skin microenvironment. Int. J. Mol. Sci. 2021, 22, doi:10.3390/ijms22083849.

  • Furthermore, when describing the properties of the different flavinoids you describe many to be favourable in both UVA and UVB damage. It is therefore necessary to be sure UVA ageing mechanisms also are mentioned in the pathophysiology of aging.

Following this valuable suggestion, a proper paragraph describing the influence of different types of radiation on the skin (including UVA-mediated photoageing) was added to the manuscript section describing the pathophysiology of skin ageing (as mentioned above)

“UV radiation plays a central role in skin senescence and skin cancer development. UV radiation is composed of three main components based on photon wavelength: UVA having the longest wavelengths (315–400 nm), UVB being mid-range (290–320 nm), and UVC being the shortest wavelengths (100–280 nm). All UV types can act as environmental mutagens leading to direct and indirect (via increased production of oxidative free radicals) DNA damage, and each can result in mutagenesis in skin cells. UVA radiation is the most prevalent component of solar UV radiation. It penetrates deeper than UVB (that has a major action on the epidermis) into the skin and induces profound alterations of the dermal connective tissue [18, 19]. In vitro studies also show that UVC has a deteriorating effect on genome stability, contributing to the aging of fibroblasts and keratinocytes [20, 21]. However, considering that most of this radiation is absorbed by an ozone layer, its clinical relevance is less pronounced. To give the complete picture, it is also important to mention the effects of infrared radiation (IR) on skin aging. Recent studies indicate that IR and heat may induce premature skin aging by stimulation of matrix metalloproteinases (MMP) expression and modulation of elastin and fibrillin synthesis. Moreover, heat stimulates in human skin formation of new vessels, recruitment of inflammatory cells, and causes oxidative DNA damage [22].” (Page 2-3, lines 80-96)

  • Table 1. You refer to reference 62. In the column ‘Disease entity’ you write: UVB induced damage. In ‘Conclusion’ column: Restoration of damage caused by UVA. Obviously, this is incorrect and the paper does not conclude with this.

We do apologize for this typo.

  • Please make sure that you site all papers correctly and that the reference actually mentions what you claim that it does. A thorough read-through for checking the validity of all the cited references seems necessary before accepting the paper. I noticed the wrongly cited papers on UVB, but all other entities and conclusions also need to be checked.

We made every effort to ensure that the citations in the revised version of the manuscript were adequately matched to the manuscript's content. All changes are marked in green.

Round 2

Reviewer 3 Report

The authors have adequately addressed all my comments. I suggest accepting the paper.